



# Subsurface Initiation of Deep Convection near Maud Rise

René M. van Westen[1] and Henk A. Dijkstra[1,2]

[1]Institute for Marine and Atmospheric research Utrecht, Department of Physics, Utrecht University, Utrecht, the Netherlands
[2]Center for Complex Systems Studies, Utrecht University, Utrecht, the Netherlands

*Correspondence to:* René van Westen <r.m.vanwesten@uu.nl>

**Abstract.** In 2016 and 2017, an open-ocean polynya appeared over Maud Rise. The formation of these polynyas has been attributed to the occurrence of intense winter storms. However, the evolution and lifetime of the two polynyas was quite different. Here, we use model output of a century long high-resolution climate model simulation to explain the differences between the 2016 and 2017 Maud Rise polynyas. Analysis of the results, using convective available potential energy to measure subsur-
face convection, leads us to the interpretation that the first polynya event is (partly) initiated by subsurface static instabilities, leading to subsurface convection. Subsurface convection associated with the formation of the 2016 polynya preconditioned the Maud Rise region, resulting in a weakly stable surface layer and eventually leading to the 2017 polynya event. Based on this, we argue that, apart from atmospheric variability, subsurface convection is important to initiate a Maud Rise polynya.

## 1 Introduction

The interest for the occurrence of Maud Rise Polynyas (MRPs) has increased recently because of the appearance of an MRP in 2016 – 2017 (Jena et al., 2019). An MRP is characterised by a sea-ice enclosed open water area near Maud Rise. The formation of a polynya over Maud Rise is a rare event and the mid-1970 MRP had the largest areal extent of $10 - 30 \times 10^4$ km$^2$ of all the observed MRPs (Gordon, 1978; Carsey, 1980; Lindsay et al., 2004; Gordon et al., 2007). The 2016 – 2017 MRP was considerable smaller compared to the mid-1970s MRP with an areal extent of $3 \times 10^4$ km$^2$ (2016) and $5 \times 10^4$ km$^2$ (2017)
(Campbell et al., 2019).

The overall mechanism of MRP formation is well known. One essential ingredient is the weakening of the ocean stratification in the Maud Rise region, also referred to as preconditioning (Martinson et al., 1981). Preconditioning can, for example, occur due to the interaction of an ocean eddy and the Maud Rise topographic feature (Holland, 2001; Jena et al., 2019) or by subsurface heat accumulation (Martin et al., 2013; Reintges et al., 2017). Subsequently, changes in atmospheric forcing lead to
the destabilisation of the water column and consequently to vertical mixing (Dufour et al., 2017). Relatively warm subsurface waters are mixed towards the surface leading to sea-ice melt and consequently MRP formation. Deep convection sustains the MRP and prevents refreezing of the sea ice. In climate model simulations, the mechanisms of subsurface heat accumulation and heat release are strongly model (and model resolution) dependent (Dufour et al., 2017; Reintges et al., 2017; Weijer et al., 2017; van Westen and Dijkstra, 2020).

In a recent study (Campbell et al., 2019) a comprehensive analysis of the 2016 – 2017 MRP is conducted. On 27 July 2016, the MRP opened after an intense winter storm and, after several weeks of quiescent storm activity, the MRP closed again on





17 August 2016. After the closing of the 2016 MRP, two (intense) winter storms struck the Maud Rise region at the end of August 2016, but these storms did not lead to any MRP formation. Surprisingly, when the 2017 MRP opened, even a longer quiescent period (1.5 months) in storm activity occurred and the MRP did not close during that year. The differences in the evolution and lifetime between the 2016 and 2017 MRP were attributed to the stronger vertical overturning in 2017 (Campbell
et al., 2019).

In this study, we aim to provide an alternative explanation for the differences between the evolution and lifetime of the 2016 and 2017 MRPs. In a simulation with a high-resolution version of the Community Earth System Model (CESM, Hurrell et al. (2013)), van Westen and Dijkstra (2020) found a multidecadal time scale preconditioning of the Maud Rise region by subsurface processes. Consequently, by deep convection, MRP formation occurred at the same multidecadal time scale, but the
formation mechanism was not discussed in van Westen and Dijkstra (2020).

Here, we analyse the formation of one of these MRPs and investigate the role of subsurface initiation of (deep) convection on polynya formation. Section 2 provides a brief overview of the climate model simulation used and information on the observational data and reanalysis products analysed. In Section 3, the formation of one multiyear MRP in CESM is analysed, together with a comparison to observations and reanalysis for the 2016 – 2017 period. A summary and discussion of the results
with the main conclusions are given in the final Section 4.

## 2   Model Output and Methods

Model output of a 250 year CESM version 1.04 control simulation is analysed. The ocean component (POP) and the sea-ice component (CICE) of the CESM have a 0.1° horizontal resolution on a curvilinear, tripolar grid which captures the development and interaction of mesoscale eddies (Hallberg, 2013). The ocean model has 42 non-equidistant depth levels, with the highest
vertical resolution near the surface. The atmosphere component and land-surface component of CESM have a horizontal resolution of 0.5°, and the atmosphere component has 30 non-equidistant pressure levels. The forcing conditions (i.e. $CO_2$, solar insulation, aerosols) are the observed ones over the year 2000 which are yearly-repeated. More details of the CESM simulation are provided in van Westen and Dijkstra (2020).

We analysed the last 101 years (model years 150 – 250) of the CESM simulation. Four (multiyear) polynyas are identified
over Maud Rise in this simulation: model years 158 – 159, 178 – 182, 205 – 209 and 231 – 237 (van Westen and Dijkstra, 2020). Here, the formation of the last multiyear MRP (model years 231 – 237) is analysed in detail, where we use both monthly-mean and daily-mean quantities. Most quantities are spatially averaged over the Polynya region (2°E – 11°E × 63.5°S – 66.5°S, see Figure 2a), which is based on the average position of all the MRPs in the CESM simulation (van Westen and Dijkstra, 2020).

The CESM results are compared to observations and reanalysis data, in particular for the period 2016 – 2017. We used
sea-ice measurements by the Scanning Multichannel Microwave Radiometer (SMMR) and Special Sensor Microwave Imager (SSM/I) (http://nsidc.org/data/G02202, Peng et al. (2013); Meier et al. (2017)). Also vertical profiles of an Argo float (5904468, http://www.coriolis.eu.org), which was deployed in 2015 and remained active till 2018, and was located near Maud Rise during the 2016 – 2017 MRP, were analysed. In addition, we retained model output from the Operational Mercator global





ocean analysis and forecast (http://marine.copernicus.eu/services-portfolio/access-to-products/) at $1/12°$ horizontal resolution. Mercator assimilates various observational data sets and the model is 'steered' towards observations. The Mercator output is stored as daily averages and the output is available from 2016 to present.

## 3  Results

In Subsection 3.1 below, we first present a detailed analysis of the last multiyear MRP event (model years $231 - 237$) in CESM, which most clearly shares phenomena as observed during the $2016 - 2017$ MRP. The details of the formation of this MRP are described in Subsection 3.2 and a comparison with observations is made in Subsection 3.3.

### 3.1  The Maud Rise Polynyas in Model Years $231 - 237$

The monthly averaged sea-ice fields (Figure 1a) show 7 consecutive years where an MRP appears, starting in September of
model year 231 and ending in November of model year 237. There is no overall decline in sea-ice fraction, $f_{ice}$, between model years $227 - 230$ (except from seasonality) and all September sea-ice fractions over this period are above the climatology mean ($\overline{f}_{\text{ice,Sep}} = 89.1$ %), which has been determined using all non-MRP years between model years $150 - 250$.

Prior to the MRP in model year 231, the subsurface (below 100 m depths) is relatively warm and salty (compared to the time mean) over Maud Rise (colour plots in Figures 1a, b). To obtain the anomalies, we averaged the temperature and salinity
($T_k$ and $S_k$, respectively) over the Polynya region (at depth $z_k$), the time-mean temperature (model years $150 - 250$) was subtracted from the time series and the result was quadratically detrended. Accumulation of subsurface heat and salt leads to preconditioning of the density in the Maud Rise region (van Westen and Dijkstra, 2020).

The MRP forms in model year 231 when the September sea-ice fraction strongly deviates from climatology ($\Delta f_{\text{ice}} = -26.1\%$), as well as for the other months during that year (Figure 1c). The mixed layer depth (black curve in Figure 1a)
also increases in model year 231, mixing positive subsurface temperature and salinity (colours in Figure 1a, b) anomalies towards the surface through convection. During the following model years ($232 - 237$), the sea-ice fractions are also below the climatological values and when convection ceases in model year 238, the sea-ice fractions return to climatology (Figure 1c).

Near the surface (upper 100 m), the temperature is seasonally varying between model years $227 - 230$. The same seasonal variability near the surface is observed in the monthly-averaged salinity, sea-ice fraction, sea-ice thickness ($H_{\text{ice}}$) fields and
in monthly maximum mixed layer depth; all quantities are averaged over the Polynya region (Figure 1a and 1b). Figure 1d shows the climatologies for $H_{\text{ice}}$ and displays similar results as the sea-ice fraction. The occurrence of an MRP strongly affects the seasonal cycle of various quantities near Maud Rise. In the following section we address the formation of this multiyear polynya event.




a) Temperature, sea-ice fraction and mixed layer

b) Salinity, sea-ice thickness and mixed layer

c) Climatology sea-ice fraction

d) Climatology sea-ice thickness

**Figure 1.** (a): Monthly-averaged sea-ice fraction ($f_{\mathrm{ice}}$), oceanic temperature anomalies and monthly maximum mixed layer depth, averaged over the Polynya region. The temperature anomalies are with respect to the time-mean (model years $150 - 250$) and are quadratically detrended. The blue dashed curve indicates the September averaged sea-ice fraction ($\overline{f}_{\mathrm{ice,Sep}}$) of all non-MRP years. The black dashed curves indicate the beginning (model year 231) and ending (model year 237) of the multiyear MRPs. (b): Similar to a), but now for the salinity anomalies, sea-ice thickness ($H_{\mathrm{ice}}$) and the September averaged sea-ice thickness ($\overline{H}_{\mathrm{ice,Sep}}$) of all non-MRP years (c): Sea-ice fraction climatology ($\overline{f}_{\mathrm{ice}}$, black curve) of all non-MRP years averaged over the Polynya region, the shading indicates the climate variability, where the dark (light) shading corresponds to the $25\% - 75\%$ ($5\% - 95\%$) percentile levels of all non-MRP years. The curves show the monthly-averaged sea-ice fractions for particular years (before, during and after MRP). (d): Similar to c), but now for the sea-ice thickness.



### 3.2 Formation of the Maud Rise Polynya

Convection is a generic aspect of the occurrence of the multiyear polynya event during model years $231 - 237$, as well as for the other polynya events (van Westen and Dijkstra, 2020). In this subsection, we analyse the initiation of vertical mixing near the Polynya region in more detail by focusing on the polynya event in model year 231.

Convection is caused by a statically unstable water column. The CESM provides a measure of this static stability (part of the standard output) through the vertical gradient of potential density $\rho_\theta$ at each depth level, i.e.,

$$\mathcal{S} = \frac{\partial \rho_\theta}{\partial z} \tag{1}$$

It turned out that the monthly-averaged CESM output is not appropriate to analyse the initiation of convection due to the strong temporal variability of this process. Therefore, we analyse daily-averaged output for the sea-ice fraction, mixed layer depth,

potential density and static stability. Due to storage limitations, these daily-averaged quantities are only available for model year 231. Recall that the MRP is defined as the enclosed region in the Antarctic sea-ice bounded by the 15% sea-ice fraction contour (Weijer et al., 2017). Prior to polynya formation in CESM (before 20 August, model year 231), the water column in the Polynya region is statically stable ($\mathcal{S} < 0$) near the surface while being statically unstable ($\mathcal{S} > 0$) at greater depths (Figures 2a, b). On 20 August, model year 231, two small (both $< 1400 \, \text{km}^2$) MRPs form and merg into a larger MRP over time (red curves

in Figure 2a, b). The daily maximum mixed layer depth (inset Figure 2a) is homogeneous and has a spatial mean of 78 m over the Polynya region.

    To investigate the role of the subsurface static instability in the polynya formation, we use the concept of Convective Available Potential Energy (CAPE), which is a measure of the potential energy available for conversion to kinetic energy of a fluid particle to move upwards or downwards in a statically unstable situation (Su et al., 2016). Using $\rho_\theta(z)$ as the background,

CAPE at depth $z$ is defined as

$$\text{CAPE}(z) = - \int\limits_{-z_{ref}}^{-z} g \frac{\rho_\theta(-z_{ref}) - \rho_\theta(z')}{\rho_\theta(z')} \, \mathrm{d}z' \tag{2}$$

where $g$ the gravitational acceleration and $\rho_\theta(z)$ the potential density of the particle at depth $z$. Starting at a reference level ($z_{ref}$) and the corresponding potential density ($\rho_\theta(z_{ref})$), we integrate upwards until CAPE becomes zero (or the surface is reached); this depth is indicated by $z_0$ and is referred to the final depth. Fluid elements which are initially located in a statically

unstable ($\mathcal{S} > 0$) layer are only considered here, since particles need to deviate from their reference depth in order to initiate convection.

    A typical vertical potential density profile in the Polynya region (Figure 2c) is statically unstable at $z_{ref} = 580$ m and, from CAPE analysis, fluid elements from this depth reach $z_0 = 29$ m. Figure 2d shows that, prior to polynya formation, subsurface convection initially at $z_{ref} = 580$ m reaches up to a maximum of $z_0 = 1$ m in the Polynya region (see inset Figure 2c). The

regions of subsurface convection for which particles reach the (near) surface overlap with the regions covered by the two small MRPs in August, model year 231.





a) Potential density gradient, 45 m

b) Potential density gradient, 480 m

c) $\rho_\theta(z)$ and final depth

d) Final depth

**Figure 2.** Static stability $\mathcal{S}$ at (a) 45 m and (b) 580 m for 17 August, model year 231. The black outlined region is the Polynya region. The red curves indicate the polynya areas of 25 August (total of $7.1 \times 10^3$ km$^2$) and 25 October ($3.6 \times 10^4$ km$^2$), model year 231. The inset in a) shows the daily maximum mixed layer depth for 17 August, model year 231. (c): Potential density ($\rho_\theta$) with depth at 4°E and 65.3°S on 17 August 231. The red (blue) region displays positive (negative) CAPE with respect to the reference potential density of 0.226 kg m$^{-3}$ at 580 m. The inset shows of the 25% highest $z_0$ (final depth) of all the potential density profiles in the Polynya region after reaching zero CAPE, the blue dashed line indicates the 5%-percentile level (i.e. $z_0 = 41$ m). (d): Final depth ($z_0$) of fluid elements initially starting at $z = 580$ m for 17 August, model year 231 due to subsurface convection.





a) Static instability, daily averages

b) Final depth

c) Static instability, monthly averages

d) Depth-averaged static instability

**Figure 3.** (a): Area per depth level where the water column in the Polynya region is statically unstable ($\mathcal{S} > 0$), normalised to the total area at that depth level. The blue curve (bottom part) is the daily-averaged magnitude of the wind-stress curl over the Polynya region, the dashed line and shading are the climatology mean and the climate variability (5 − 95%-percentile level), respectively, derived from all the monthly averages. The wind-stress curl intervals are spaced by $10 \times 10^{-7}$ N m$^{-3}$. (b): Final depth of fluid elements due to (subsurface) convection, using the 5%-percentile level of the potential density profiles in the Polynya region (see inset Figure 2c). Only regions where deviations from the initial depth occur are displayed. The red curve shows the final depth of convection initially starting at $z_{ref} = 580$ m. The blue curve (bottom part) is the total area of the model year 231 MRP. The dashed lines in a) and b) indicate the formation (20 August) and ending (14 December) of the MRP. The black curve shows the daily maximum mixed layer depth averaged over the Polynya region. (c): Same as a), but now for monthly-averaged values of the static stability. The black curve is the monthly maximum mixed layer depth, averaged over the Polynya region. (d): Normalised area of static instability averaged over the upper 100 m and between 200 − 1000 m. The dashed lines in c) and d) indicate the beginning (model year 231) and ending (model year 237) of the multiyear MRPs.





The water column is also statically unstable at other depths than $z_{ref} = 580$ m (Figure 3a). Following the same analysis as above, we determined the final depth (i.e. $z_0$) for all depth levels which are statically unstable in the Polynya region. Since subsurface convection is (very) localised (Figure 2d), we retained the 5%-percentile level of the final depth of all potential density profiles in the Polynya region as a measure for the magnitude of subsurface convection of the Polynya region. For

17 August, model year 231, for example, convection initially at $z_{ref} = 580$ m results in a final depth of 41 m at the 5%-percentile level (see inset Figure 2c). The time evolution of the 5%-percentile final depth for all vertical layers is shown in Figure 3b, where the red curve shows the final depth for one particular reference level (i.e. $z_{ref} = 580$ m). Subsurface convection is initiated before (mid – end of July) the formation of the MRP on 20 August, model year 231 (see blue curve in Figure 3b). Subsurface convection almost overcomes the stratification near the surface at the 5%-percentile level of the final

depth; selecting a lower percentile level ($< 3\%$) reveals that the subsurface convection reaches the surface (not shown).

The static instabilities of the Polynya region over a longer period of time, using monthly-averaged output, are shown in Figure 3c. The monthly-averaged static instabilities are weaker compared to the daily-averaged ones (Figure 3a) for model year 231. Despite the underestimation of the monthly-averaged static instabilities over the Polynya region, the subsurface (200 – 1000 m) is also getting more statically unstable between model years 227 – 231, prior to polynya formation (Figure 3d). The

surface layer (upper 100 m) is seasonally varying during model years 227 – 231 and the surface static instability is related to cooling of the surface layer and to brine rejection by sea-ice formation during austral winter. Surface static instabilities are also reflected in the monthly maximum mixed layer depth, which displays the same seasonal variability. The CAPE analysis, using monthly-averaged potential density and static instability profiles, hardly shows any subsurface convection (not shown) compared to the daily-averaged output (Figure 3b). Hence, daily-averaged output is required to analyse the effect of subsurface

convection on polynya formation.

Subsurface convection favours MRP formation, as relatively warm and saline water is mixed towards the surface. During the 2016 MRP, however, the sea-ice near Maud Rise opened up by intense winter storms and the storm frequency and intensity was linked to a positive phase of Southern Annular Mode (SAM) index (Campbell et al., 2019). In the CESM, the model year 231 – 237 MRP is also preceded by a positive SAM index, as well as for the second (178 – 182) and third (205 - 209)

multiyear MRPs (not shown). In the CESM, the SAM index is also significantly (as in observations (Campbell et al., 2019)) anti-correlated ($r = -0.63$) with the sea-level pressure over the Polynya region. The sea-level pressure over the Polynya region is in turn significantly correlated ($r = 0.56$) with the wind-stress curl averaged over the Polynya region. A negative wind-stress curl leads to sea-ice divergences, upwelling (by Ekman dynamics) and entrains relatively warm and saline subsurface water towards the surface, potentially leading to MRP formation.

Therefore, we determined the magnitude of the daily-averaged wind stress curl over the Polynya region for model year 231. From the monthly averages we determined the climatologies over all the model years, and both are shown in bottom panel of Figure 3a. Prior and during MRP formation (17 – 24 August, model year 231), the wind-stress curl is positive over the Polynya region leading to Ekman downwelling over the Polynya region. A positive wind-stress curl induces sea-ice convergences. On 16 August, model year 231, a few days before the polynya emerges, relatively large negative wind-stress curl values are found

over the Polynya region. These negative wind-stress curl values are mainly situated in the northern part of the Polynya region





(not shown), outside of the regions where the two polynyas form (cf. Figures 2a, b), but still they enhance upwelling, turbulent mixing and sea-ice divergences and hence can favour MRP formation.

After (deep) convection is initiated which vertically mixes heat and salt upwards, the surface water in the Polynya region remains relatively warm and saline compared to the time mean (Figure 1a, b). The relatively warm surface water slows down

and prevents the formation of sea ice near Maud Rise during the following year (model year 232, Figure 1a). The open-ocean water area in the Polynya region is strongly cooled during austral winter, which decreases the static stability of the water column near the surface. Deep convection also mixes subsurface salinity anomalies towards the surface which decrease the static stability near the surface. The combination of positive temperature and salinity anomalies result in surface driven convection which is shown in Figure 3d by the relatively large surface instabilities after model year 231. After several years,

the subsurface heat reservoir is depleted and convection ceases, leading to a statically stable water column.

### 3.3 Subsurface Convection in Observations and Mercator Data

In this subsection, we analyse if there was any subsurface convection during the 2016 – 2017 MRP. The analysis consists of measured sea-ice fractions in SSMR-SSM/I data, Argo float measurements and Mercator model output (as discussed in section 2).

First, we determined the daily sea-ice fraction averaged over the Polynya region between 1990 – 2017 from the SSMR-SSM/I (Merged sea-ice concentrations). The sea-ice fraction climatology over 1990 – 2015 and the sea-ice fractions in 2016 and 2017 are shown in Figure 4a. In August 2016 during the opening of the sea ice (27 July – 17 August 2016), relatively low sea-ice fractions were measured with respect to the climatology over the Polynya region. On 5 August 2016, for example, the measured sea-ice fraction over the Polynya region was $f_{\text{ice}} = 65\%$ ($\overline{f}_{\text{ice,5Aug}} = 97\% \pm 2\%$, 1990 – 2015). The following

year, an MRP formed in September 2017 and relatively low sea-ice fractions (with respect to climatology) were observed over the Polynya region between September and December. On 16 September 2017, the sea-ice fraction was $f_{\text{ice}} = 58\%$ ($\overline{f}_{\text{ice,16Sep}} = 97\% \pm 2\%$, 1990 – 2015). Note that the CESM model results (also plotted in Figure 4a) display comparable sea-ice fractions (averaged over the Polynya region) of about $f_{\text{ice}} = 60\%$ when the MRP formed in model year 231 (cf. Figure 1c).

Second, the oceanic state between 2016 – 2017 is analysed using model output from Mercator, which provides daily averages

of oceanic potential temperature, salinity, mixed layer depth and sea-ice fraction. The Mercator sea-ice fractions (averaged over the Polynya region) in 2016 and 2017 are also displayed in Figure 4a and reasonably agree with observations. The vertical profiles of potential temperature and salinity are shown in Figure 4b and compared to those of an Argo float (5904468) that was present near Maud Rise in 2016 – 2017. Although these Argo float observations are too sparse (every 10 days) to analyse the oceanic state (and e.g. convection), they can be used to validate the Mercator data which has a similar vertical background

profile as observations.

Applying the same CAPE analysis as in section 3.2 to the Mercator data, we determined the final depth of (subsurface) convection in the Polynya region. Because the static stability is not provided in the Mercator data, it was assumed that layer $\text{d}z_k$ is statically unstable ($\mathcal{S}_k > 0$) when $\rho_\theta(z_{k+1}) < \rho_\theta(z_{k-1})$, where index $k$ increases with increasing depth. Next, from the



a) Sea-ice fraction

b) Potential temperature & Salinity

c) Final depth

d) Final depth time evolution

**Figure 4.** (a): Daily sea-ice fractions averaged over the Polynya region, for the SSMR-SSM/I, Mercator and CESM model year 231. The black curve is the sea-ice fraction climatology ($\overline{f}_{ice}$, $1990 - 2015$) for SSMR-SSM/I. (b): Potential temperature and salinity profiles of Mercator and the Argo float at $4.5°$E and $65.7°$S on 8 August 2016 (white star in c)). (c): Final depth of subsurface convection initially starting at $z = 1940$ m and the inset shows the mixed layer depth on 25 July 2016. The blue curves indicate the relatively low sea-ice fractions contour of $75\%$ on 4 August 2016. (d): Final depth of (subsurface) convection, using the $5\%$-percentile level of the Mercator potential density profiles in the Polynya region. Only regions where deviations from the initial depth occur are displayed. The red curve shows the final depth of convection for $z_{ref} = 1940$ m. The black curve shows the $95\%$-percentile level of daily mixed layer depth of the Polynya region. The dashed lines indicate the formation and ending of the MRP in 2016 and 2017.





vertical profiles of potential density and static instability, the final depth of (subsurface) convection was determined using the expression for CAPE in (2).

In the Polynya region, there are several periods between 2016 – 2017 where the subsurface is statically unstable, in particular for July and August 2016 (Figure 4c). For example, on 25 July 2016 (two days before polynya formation), deep convection

initially at $z_{ref} = 1940$ m overcomes the stratification near the surface. The 75% Mercator sea-ice fraction contours (blue curves in Figure 4c) partly overlap with the regions of (deep) convection. This indicates that subsurface heat is vertically mixed towards the surface leading to relatively low sea-ice fractions over the Polynya region (Figure 4a). There is no surface driven mixing prior to polynya formation (inset Figure 4c), as the mixed layer depth is homogeneous and shallow (mean of 77 m) over the Polynya region. The Argo float is not located in the regions of the Mercator subsurface convection (also tested for other

days). However, there are subsurface static instabilities in the Argo profiles. From the profiles in Figure 4b, for example, this results in convection starting at $z_{ref} = 240$ m and reaching up to $z_0 = 110$ m. This (limited) subsurface convection explains the vertically homogeneous potential temperature and salinity profiles between 110 – 240 m depths in the Argo (and also Mercator) potential density profiles (Figure 4b).

The final depth of (subsurface) convection for the 5%-percentile level of the Mercator potential density profiles in the

Polynya region is shown in Figure 4d for the period 2016 – 2017. Between June and early August 2016, subsurface convection is initiated and overcomes the stratification near the surface. The combination of intense winter storms (Campbell et al., 2019) and subsurface convection leads to polynya formation. After the polynya closes on 17 August 2016, the intensity of subsurface convection decreases, as seen by the increase in final depth values in Figure 4d. This explains why the polynya did not reappear in late August 2016 when two intense winter storms struck the Maud Rise region (Campbell et al., 2019). The sea-ice fractions

over the Polynya region return to climatological values in the absence of subsurface convection (Figure 4a).

The formation of the 2017 MRP is strongly influenced by the 2016 MRP which preconditioned the density field in the Polynya region. Before the 2017 MRP, the upper-ocean had low static stabilities and deep convection was much better developed compared to the period before the 2016 MRP (Campbell et al., 2019). The mixed layer depth was indeed deeper in 2017 compared to 2016 (Figure 4d) and locally the mixed layer depth reached up to 500 m and 1200 m depths in 2016 and 2017,

respectively. In Figure 4d, the displayed mixed layer depth is the 95%-percentile level of the daily mixed layer depth of the Polynya region; the deepening of the mixed layer depth is not visible when taking the average over the Polynya region. Subsurface convection did not overcome the stratification near the surface in 2017 (also tested at the 1%-percentile level). Hence, the 2017 MRP was mainly caused by surface static instabilities rather than subsurface convection.

When comparing Mercator results with those in CESM for model year 231, subsurface convection is active prior to polynya

formation in both models. In Mercator, subsurface convection has a much larger vertical extent (up to 2000 m) compared to the CESM (up to 1000 m). In both models, subsurface convection overcomes the stratification near the surface leading to a decrease in sea-ice fraction over Maud Rise. After the first MRP (i.e. year 2016 and model year 231) preconditioned the Polynya region, the second MRP (i.e. year 2017 and model year 232) formed by surface instabilities (Figure 3d). If subsurface convection would have remained active for a longer period of time in 2016, it could have prevented the closing of the 2016 MRP.





## 4    Summary and Discussion

In this paper, we analysed model output from a multi-centennial (250 years) control simulation of a high-resolution version of the Community Earth System Model (CESM) under a repeated seasonal forcing of the year 2000. In the last 100 years of the simulation, four multiyear polynyas formed over Maud Rise (van Westen and Dijkstra, 2020) and here we analysed the

formation of the last polynya event (model years 231 – 237). A layer between 200 – 1000 m depths is statically unstable, leading to subsurface convection which causes vertical mixing of positive temperature and salt anomalies towards the surface. The anomalous heat melts sea ice and leads to polynya formation in model year 231, following favourable wind-stress conditions. The surface layer (upper 100 m) remains statically unstable and initiates the next polynya event in model year 232 through atmospheric forcing.

In the Mercator data for the 2016 – 2017 period, we also find subsurface convection near Maud Rise. In 2016, subsurface convection led to negative sea-ice anomalies (with respect to climatology) over Maud Rise and in combination with an intense winter storm (Campbell et al., 2019), a polynya formed. Subsurface convection decreased after the formation of the 2016 MRP, which explains the relatively short lifetime (3 weeks) of this polynya. The surface layer near Maud Rise remained weakly stratified after the 2016 MRP and surface static instabilities (in combination with atmospheric variability) initiate the 2017 MRP,

similar to the results in the CESM simulation.

From these results we conclude that subsurface convection has played a significant role in the 2016 MRP development. Although both the 2016 and 2017 MRPs are likely initiated by intense winter storms (Campbell et al., 2019; Jena et al., 2019), the presence of the subsurface heat reservoir provided a sufficient preconditioning of the density field to allow subsurface convection to reach the surface and to melt the sea ice. Furthermore, the lifetime of the 2016 MRP was determined by the

intensity of subsurface convection. When subsurface convection ceased (Figure 4d), the sea ice did refreeze over Maud Rise (Figure 4a). After the region was preconditioned by the 2016 MRP, the surface static instabilities and surface driven convection determined the lifetime of the 2017 MRP. Subsurface convection did not overcome the stratification near the surface in 2017 and hence played no role in its lifetime.

The role of subsurface heat reservoir in the development of MRPs is interesting considering the CESM results in van Westen

and Dijkstra (2020), where a multidecadal ($\sim$ 25 year) dominant time scale of preconditioning was found over Maud Rise. This time scale originated from the variability associated with the Southern Ocean Mode (Le Bars et al., 2016) through propagation of subsurface heat anomalies along the Weddell Gyre. Within this view, the relatively low sea-ice concentrations measured near Maud Rise in 1994 (Lindsay et al., 2004; Gordon et al., 2007) possibly were related to subsurface convection (similar to the 2016 MRP), which did not overcome the stratification at the surface. It would also explain why so few MRPs have developed

over the last decades in spite of the highly variable atmospheric forcing conditions over Maud Rise.

*Acknowledgements.* The authors thank Michael Kliphuis (IMAU, UU) for performing the CESM simulations on the Cartesius at SURF-sara in Amsterdam within project 15552. The data from the model simulation used in this work are available upon reasonable request from the authors. The Argo data were collected and made freely available by the Coriolis project and programs that contribute to it



(http://www.coriolis.eu.org). The NOAA/NSIDC provided the satellite sea-ice products (http://nsidc.org/data/G02202). The Mercator data set was provided by Copernicus-Marine environment monitoring service (http://marine.copernicus.eu/services-portfolio/access-to-products/).



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
