# Peer review of "Subsurface Initiation of Deep Convection near Maud Rise"

_Ocean Science, 2020_

## Referee Comment (RC1) · Anonymous Referee #1 · 26 Jun 2020

Subsurface Initiation of Deep Convection near Maud Rise by René M. van Westen and Henk A. Dijkstra

Description: Model output from a 250-year high-resolution CESM simulation is analyzed to study Maud Rise Polynyas in the simulation. The authors try to use the Maud Rise Polynyas in the simulation to explain the difference between the observed Maud Rise Polynya events in 2016 and 2017 which were triggered by intense winter storms and offer an alternative explanation which is that subsurface convection triggers a Maud Rise Polynya.

The manuscript addresses some interesting questions - what is the preconditioning and triggering mechanisms of Maud Rise polynyas? Given the observed (2016-2017) polynyas over Maud Rise, it would be interesting to know why these did not lead to

a larger Weddell Sea Polynya. One of the obvious limitations in the study is that the authors do not define what subsurface convection is and do not provide a clear justification for using the Convective Available Potential Energy (CAPE) to indicate the onset of convective overturning in the Weddell Sea. They also do not investigate several previously suggested significant preconditioning processes that are relevant to the Maud Rise and the Weddell Sea, including the size of the sub-surface heat reservoir created by accumulation of Warm Deep Water, the strength of the Taylor column associated with the Maud Rise seamount, the strength of the impinging flow, and surface salinity anomalies. While I think this is an interesting problem, and the authors have looked into some of the possible processes, I am not convinced by the robustness of the results and do not suggest publishing at this time. There is a lot of inconsistency in the way the authors define Convective Available Potential Energy (CAPE) and also use results by Su et al (2016) to justify that CAPE indicates the onset of convection.

Reviewer Decision: I do not recommend publishing the article.

Major Issues:

1. In section 3.1, the authors show the accumulation of subsurface heat and salt prior to the onset of the polynyas but fail to relate it to the heat and salt content of a warm and salty water mass known as the Weddell Deep Water (WDW) which lies between 250-1500m in the Weddell Sea. A necessary but not sufficient condition for long-lasting, consecutive winter polynyas to occur is the heat reservoir at depth or the heat content of WDW (Martinson et al. 1981; Martin et al. 2013; Cheon et al. 2015; Dufour et al. 2017; Kurtakoti et al. 2018). If the main result in the paper is that the convection is initiated subsurface, a thorough and robust analysis of the stratification and water masses in and around Maud Rise is critical to the authors' justification and the readers' understanding.

2. Please define what is subsurface convection. How is it triggered and how is it maintained? Is this convection that only occurs within the WDW and if so how is it

initiated and how is this sustained?

3. One of the major scientific weaknesses of this study is the context in which CAPE is used. Ocean CAPE is defined for an ocean column as the maximal Potential Energy that can be released by adiabatic vertical parcel rearrangements, arising from thermobaricity (Su et al. 2016a,b). However, that alone does not indicate the onset of deep convection. Fig 3a. is used to indicate the onset of subsurface convection in early- mid July which is not accurate. Fig 3c clearly shows that the subsurface pattern repeats itself in the 2 years prior to the formation of the polynyas. This only indicated weakening subsurface stratification and not the onset of subsurface convection which is consistent with the accumulation of WDW heat content. If we could also see the Hovmöller of Temperature and Salinity time series that overlaps the time periods used in 3a and 3c, we can then confidently know for sure if convection does begin in the subsurface as claimed by the authors.

References

Cheon, W. G., S. Lee, A. L. Gordon, Y. Liu, C. Cho, and J. J. Park, 2015: Replicating the 1970s' Weddell Polynya using a coupled ocean-sea ice model with reanalysis surface flux fields. Geophys. Res. Lett., 42, 5411–5418, doi:10.1002/2015GL064364.

Dufour, C. O., A. K. Morrison, S. M. Griffies, I. Frenger, H. Zanowski, and M. Winton, 2017: Preconditioning of the Weddell Sea polynya by the ocean mesoscale and dense water overflows. J. Clim., 7719–7737, doi:10.1175/JCLI-D-16-0586.1. http://journals.ametsoc.org/doi/10.1175/JCLI-D-16-0586.1.

Kurtakoti, P., M. Veneziani, A. Stössel, and W. Weijer, 2018: Preconditioning and Formation of Maud Rise Polynyas in a High-Resolution Earth System Model. J. Clim., 31, 9659–9678, doi:10.1175/JCLI-D-18-0392.1. http://journals.ametsoc.org/doi/10.1175/JCLI-D-18-0392.1.

Martin, T., W. Park, and M. Latif, 2013: Multi-centennial variability controlled by

[Figure]

Southern Ocean convection in the Kiel Climate Model. Clim. Dyn., 40, 2005–2022, doi:10.1007/s00382-012-1586-7.

Martinson, D. G., P. D. Killworth, and A. L. Gordon, 1981: A convective model for the Weddell Polynya. J. Phys. Oceanogr., 11, 466–488, doi:10.1175/1520-0485(1981)011. http://journals.ametsoc.org/doi/abs/10.1175/1520-0485(1981)011%3C0466:ACMFTW%3E2.0.CO;2.

Su, Z., A. P. Ingersoll, A. L. Stewart, and A. F. Thompson, 2016a: Ocean convective available potential energy. Part I: Concept and Calculation. J. Phys. Oceanogr., 46, 1081–1096, doi:10.1175/JPO-D-14-0155.1.

Su, Z., A. P. Ingersoll, A. L. Stewart, and A. F. Thompson, 2016b: Ocean convective available potential energy. Part II: Energetics of Thermobaric Convection and Thermobaric Cabbeling. J. Phys. Oceanogr., 46, 1097–1115, doi:10.1175/JPO-D-14-0156.1.

---

## Short Comment (SC1) · 28 Jun 2020

I appreciate the authors' submission of this interesting work to Ocean Science. The journal offers members of the broader scientific community the opportunity to comment on the preprint during the eight-week open discussion period. As the lead author of Campbell et al. (2019), a study also addressing the formation mechanisms of the 2016 and 2017 Maud Rise polynyas, which is cited (and challenged) throughout this work, I feel compelled to offer an observationally-grounded perspective on some key aspects of this study. My comments below are not meant to be an exhaustive review, but I hope that the authors and editor find them useful.

The primary conclusion of this study is that the 2016 Maud Rise polynya was initiated

by subsurface overturning, which the authors identify through the appearance of deep potential density (sigma-theta) inversions in a 1/10° CESM control run (with polynya years picked to represent an analogue for 2016-17 conditions) as well as Mercator ocean reanalysis fields.

Simply put, the simulated phenomenon that is the focus of this study has not been observed in the real ocean, to my knowledge, and is in direct conflict with decades of hydrographic observations from the Weddell Sea, including profiling float measurements collected before, during, and after the 2016-2017 polynyas.

First, the cause of the subsurface destratification seen in the authors' CESM simulation is a multidecadal buildup of heat in mid-depth layers (200-1000 m) of the Weddell Sea, as detailed in their companion submission to Ocean Science Discussions (van Westen and Dijkstra, in review). Similar periodic behavior has been documented in other climate models on various timescales (Behrens et al. 2016; Reintges et al. 2017). While many in the modeling community regard this behavior as spurious and unrealistic (e.g., Heuzé et al. 2015; Held et al. 2019), I agree with the authors' view that the underlying cause of these cyclical, heat-accumulation-induced polynya occurrences is worth understanding. These cycles certainly have an outsized impact on intrinsic global climate variability in some models, as the authors and others have demonstrated.

However, oceanographic observations do not support this behavior. Repeat hydrographic surveys from 1984-2008 along the Greenwich Meridian, crossing Maud Rise, show no sign of marked heat accumulation following ventilation during the massive 1974-1976 Weddell polynyas (Fahrbach et al. 2011). The small warming trend observed is over one order of magnitude smaller than that examined in the authors' companion submission following polynya events (van Westen and Dijkstra, in review), and large decadal fluctuations complicate the detection of even that small observed trend. In fact, local rebound in heat content near Maud Rise following the major 1974-1976 polynyas occurred by 1984 at the latest (Smedsrud 2005), and likely in shorter time. Analysis of about 3,000 temperature profiles from 2002-2017 from ships, floats, and in-

strumented seals indicated that there was no local buildup of mid-depth heat leading to the 2016-2017 Maud Rise polynyas (Campbell et al. 2019; see section 'Sub-pycnocline temperature records' and Extended Data Fig. 9). These substantial differences between models and reality unfortunately limit the utility of the authors' CESM simulation as a direct analogue for investigating the recent polynya events.

In contrast, the published high-resolution model simulations that best reproduce the Maud Rise polynya phenomenon point to upper-ocean destratification from surface salinity anomalies, not subsurface heat accumulation, as most important in triggering polynyas (Kurtakoti et al. 2018; Kaufman et al. 2020). These papers should be cited and discussed. Along similar lines, the authors neglect the observational and theoretical evidence that points to low upper-ocean haline stratification as a critical factor in allowing Maud Rise polynyas to emerge in 2016 and 2017 but not in most other years (Campbell et al. 2019). It is inaccurate to characterize our study as attributing the 2016 polynya solely to intense winter storms (as stated in the authors' Abstract, Lines 1-2; Page 8, Lines 21-23; Page 12, Lines 11-12). Both weak upper-ocean stratification and strong storms appear to have been necessary, and strong storms – which were more frequent than usual in 2016 but are still a regular occurrence in most (non-polynya) years – are apparently not a sufficient condition for polynya formation. While the authors attribute the 2017 polynya to a "weakly stable surface layer" (e.g., Abstract, Line 7), they look elsewhere for the immediate cause of the earlier 2016 polynya. I find this puzzling, as the profiling float measurements examined in Campbell et al. (2019) show that a "weakly stable surface layer" existed both prior to the 2016 polynya as well as during the following winter.

Surface-driven deep convection is a phenomenon that has been well-documented in decades of observations from the North Atlantic and Mediterranean Sea (e.g., de Jong et al. 2012; Testor et al. 2018). Its theoretical underpinnings are also reasonably well-understood (e.g., Marshall and Schott 1999). In my view, there is little reason to expect that deep convection in the Weddell Sea is not similarly initiated by buoyancy

loss and/or dynamic perturbations in the upper ocean. Indeed, this is the canonical model for the formation of polynyas near Maud Rise (Martinson et al. 1981; Motoi et al. 1987). However, if the authors wish to assert that subsurface-initiated deep convection is of real importance, I would suggest critically engaging with literature that may offer a theoretical basis for this mechanism. Harcourt (2005) and Akitomo (2019), for instance, describe subtle processes by which subsurface overturning may be initiated through nonlinearities in the seawater equation of state, but it is important to recognize that this is a fundamentally different sequence of events than that which seems to be occurring in the authors' CESM run.

Regardless of the theoretical feasibility of subsurface-initiated overturning, the phenomenon that the authors highlight from their CESM run is unlikely to have occurred in 2016. In the attached Figure 1 (see below), I have plotted potential density profiles from the Argo float 5904471, which was present at Maud Rise during the recent polynyas (Campbell et al. 2019). This figure is a direct comparison to the authors' Figure 2c from their CESM run, which shows a statically unstable profile below about 75 m prior to polynya formation. In contrast, the observations show that potential density increased steadily with depth, including immediately prior to the 2016 polynya, as is typical throughout the world ocean. While the CESM profile seems to be characterized by a bolus of mid-depth heat anomalies, which create a remarkably thick potential density inversion layer, these features are unsurprisingly absent in the observations. (This is to say nothing of the model's twofold bias in its deep-to-surface potential density difference, which is greater than 0.2 kg/m3 in the model but is no more than 0.1 kg/m3 in the observations, or the inappropriate use of surface-referenced potential density [sigma-theta] to characterize inversions when a locally-referenced potential density should be used instead.)

The authors seem to dismiss the possibility of using the float observations to conduct an analysis similar to their assessment of CESM model output and Mercator reanalysis data, stating that "these Argo float observations are too sparse (every 10 days) to

analyse the oceanic state (and e.g. convection)" (Page 9, Lines 28-30). In the context of investigating preconditioning for a polynya, this is inaccurate. The oceanic state – particularly below the mixed layer – does not vary substantially on time scales less than 10 days, as seems to be indicated by the authors' own analysis of the CESM output and Mercator data. The float data are plenty useful and are, in fact, the best records we have on conditions in 2016 and 2017 at Maud Rise. (It is important to note here that the Mercator reanalysis data is poorly constrained in the ice-covered Weddell Sea, and should be approached with greater caution than is done here. Comparison with a single float profile (Figure 4b) – towards which the reanalysis is likely nudged – is not an adequate validation of its skill.)

Ultimately, the float observations analyzed in Campbell et al. (2019) as well as the body of previous work on Weddell Sea polynyas and hydrography offer ample evidence that model-observation disagreement is severe in this region and in the context of this phenomenon. With this in mind, I would argue that it is probably counterproductive to interrogate the causes of observed polynya events through direct comparison with a model that behaves very differently than the real world.

References

Akitomo, K., 2019: Stability and slow overturning of the water column in the Weddell Sea under a sea ice cover. J. Oceanogr., 75, 95–109, doi:10.1007/s10872-018-0488-7.

Behrens, E., G. Rickard, O. Morgenstern, T. Martin, A. Osprey, and M. Joshi, 2016: Southern Ocean deep convection in global climate models: A driver for variability of subpolar gyres and Drake Passage transport on decadal timescales. J. Geophys. Res. Ocean., 121, 3905–3925, doi:10.1002/2015JC011286.

Campbell, E. C., E. A. Wilson, G. W. K. Moore, S. C. Riser, C. E. Brayton, M. R. Mazloff, and L. D. Talley, 2019: Antarctic offshore polynyas linked to Southern Hemisphere climate anomalies. Nature, 570, 319–325, doi:10.1038/s41586-019-1294-0.

Fahrbach, E., M. Hoppema, G. Rohardt, O. Boebel, O. Klatt, and A. Wisotzki, 2011: Warming of deep and abyssal water masses along the Greenwich meridian on decadal time scales: The Weddell gyre as a heat buffer. Deep Sea Res. Part II Top. Stud. Oceanogr., 58, 2509–2523, doi:10.1016/j.dsr2.2011.06.007.

Harcourt, R. R., 2005: Thermobaric cabbeling over Maud Rise: Theory and large eddy simulation. Prog. Oceanogr., 67, 186–244, doi:10.1016/j.pocean.2004.12.001.

Held, I. M., and Coauthors, 2019: Structure and performance of GFDL's CM4.0 climate model. J. Adv. Model. Earth Syst., 11, 3691–3727, doi:10.1029/2019MS001829.

Heuzé, C., J. K. Ridley, D. Calvert, D. P. Stevens, and K. J. Heywood, 2015: Increasing vertical mixing to reduce Southern Ocean deep convection in NEMO3.4. Geosci. Model Dev., 8, 3119–3130, doi:10.5194/gmd-8-3119-2015.

de Jong, M. F., H. M. van Aken, K. Våge, and R. S. Pickart, 2012: Convective mixing in the central Irminger Sea: 2002–2010. Deep Sea Res. Part I Oceanogr. Res. Pap., 63, 36–51, doi:10.1016/j.dsr.2012.01.003.

Kaufman, Z. S., N. Feldl, W. Weijer, and M. Veneziani, 2020: Causal interactions between Southern Ocean polynyas and high-latitude atmosphere-ocean variability. J. Clim., 33, 4891–4905, doi:10.1175/JCLI-D-19-0525.1.

Kurtakoti, P., M. Veneziani, A. Stössel, and W. Weijer, 2018: Preconditioning and formation of Maud Rise polynyas in a high-resolution earth system model. J. Clim., 31, 9659–9678, doi:10.1175/JCLI-D-18-0392.1.

Marshall, J., and F. Schott, 1999: Open-ocean convection: Observations, theory, and models. Rev. Geophys., 37, 1–64, doi:10.1029/98RG02739.

Martinson, D. G., P. D. Killworth, and A. L. Gordon, 1981: A convective model for the Weddell Polynya. J. Phys. Oceanogr., 11, 466–488, doi:10.1175/1520-0485(1981)011<0466:ACMFTW>2.0.CO;2.
Motoi, T., N. Ono, and M. Wakatsuchi, 1987: A mechanism for the formation of the Weddell Polynya in 1974. J. Phys. Oceanogr., 17, 2241–2247, doi:10.1175/1520-0485(1987)017<2241:AMFTFO>2.0.CO;2.

Reintges, A., T. Martin, M. Latif, and W. Park, 2017: Physical controls of Southern Ocean deep-convection variability in CMIP5 models and the Kiel Climate Model. Geophys. Res. Lett., 44, 6951–6958, doi:10.1002/2017GL074087.

Smedsrud, L. H., 2005: Warming of the deep water in the Weddell Sea along the Greenwich meridian: 1977–2001. Deep Sea Res. Part I Oceanogr. Res. Pap., 52, 241–258, doi:10.1016/j.dsr.2004.10.004.

Testor, P., and Coauthors, 2018: Multiscale observations of deep convection in the northwestern Mediterranean Sea during winter 2012-2013 using multiple platforms. J. Geophys. Res. Ocean., 123, 1745–1776, doi:10.1002/2016JC012671.

van Westen, R. M., and H. A. Dijkstra, in review: Multidecadal preconditioning of the Maud Rise Polynya region. Ocean Sci. Discuss., doi:10.5194/os-2020-25.
* * *
[Figure]

[Figure]

**Profiles from Argo float 5904471 near Maud Rise**

Legend:
- 2016-07-10 (2.5°E, 64.9°S)
- 2016-08-20 (2.5°E, 65.0°S)
- 2016-09-20 (2.5°E, 65.0°S)
- 2017-07-06 (4.5°E, 64.9°S)
- 2017-08-16 (4.8°E, 64.8°S)
- 2017-09-16 (5.1°E, 64.8°S)

X-axis: Potential density with respect to surface (kg m$^{-3}$)

Y-axis: Depth (m)

**Fig. 1.** Float observations presented as a point of comparison to the authors' Figure 2c.

---

## Referee Comment (RC2) · Anonymous Referee #2 · 22 Aug 2020

General Comments:

This study investigates the initiation of deep convection over Maud Rise seamount, as well as the resultant open-ocean polynyas that appear in the region during austral winter. The authors highlight key oceanographic processes underlying the recently observed 2016-2017 polynya event, comparing high-resolution model output with observation-based data. This research topic certainly deserves attention; the seemingly rare modern occurrence of Southern Ocean deep convection, as well as its role in the high-latitude climate system, is still not fully understood. This study argues that deep convection is partially initiated by subsurface static instabilities, an explanation that contrasts from the surface-based triggering mechanisms identified

in prior high-resolution modeling studies (Kurtakoti et al., 2018) and observations (Cheon and Gordon, 2019; Campbell et al., 2019). To support this novel interpretation, the authors show that polynya formation is preceded by a weakening subsurface stratification in both CESM and Mercator model output. I found these analyses interesting, but I have two primary concerns with the study's conclusions as currently presented. First, it is not clear from the results that subsurface instabilities are a robust cause of deep convection. Concurrent changes in near-surface salinity and temperature, driven by atmospheric variability, may be the more dominant causal mechanism. Second, more attention should be given to temporal variability in the subsurface heat reservoir, given the outsized Weddell Deep Water (WDW) warming trends seen in some climate models. Acknowledging this potential model bias is especially important when considering the preconditioning mechanisms for Maud Rise polynya formation. I further detail each of these topics in my specific comments and suggestions below. If these issues are sufficiently addressed with additional analyses, I believe the model-to-observation comparison conducted in this study can provide substantial insights. Accordingly, I recommend this paper be published after major revisions are made.

Specific Comments:

1. The growing subsurface instabilities that precede the polynya in CESM model year 231 (Fig. 3a, 3b) are intriguing. It would be great if the authors could discuss in further detail what mechanism is behind the growing instability. Is it related to the multidecadal buildup of subsurface heat, mentioned in pg. 12, L24-25? Or is a shorter-timescale process more relevant here? Pg. 8 would be a good place to discuss this.

2. The aforementioned subsurface instabilities also appear to occur in the model years preceding year 231 (Fig. 3c). If these instabilities play a role in initiating

convection, why are there no polynyas during this previous time period? Are unfavorable near-surface conditions inhibiting deep convection from above? The time series of wind stress curl, as shown in Fig. 3a, could be useful if it is extended to include this earlier time period.

3. Pg. 8, L28-29 and pg. 9, L1-2: Wind stress curl is associated with upwelling, turbulent mixing, and sea-ice divergences. The manuscript would benefit from the authors explicitly quantifying the upwelling magnitude associated with wind stress curl anomalies. For instance, the horizontal and/or vertical Ekman velocities can be inferred from wind stress curl (e.g. Campbell et al., 2019, Methods, salinity fluxes from upwelling). This quantity could help contextualize the near-surface destratification shown in Fig. 3.

4. When comparing subsurface convection in CESM and Mercator output (Pg. 11, L29-35), it is important to acknowledge the magnitude of ocean heat content variability in the models used. Climate models are known to be prone to excessive subsurface heat accumulation, which has been attributed to freshwater forcing biases (Stössel et al., 2015) and weak parameterized mixing under sea ice (Heuzé et al., 2013). For instance, are subsurface warming trends between the simulated polynya events in CESM (Pg. 2, L24-25) consistent with the observed .032 K/decade trend in Weddell Deep Water temperature between 1977-2001 (Smedsrud, 2005)? What about the Mercator output? If not, the model-to-observation comparison could still be useful, but the difference must be acknowledged to properly inform the interpretation of the data.

References:

1. Campbell, Ethan C., et al. "Antarctic offshore polynyas linked to Southern Hemisphere climate anomalies." Nature 570.7761 (2019): 319-325.

2. Cheon, Woo Geun, and Arnold L. Gordon. "Open-ocean polynyas and deep convection in the Southern Ocean." Scientific reports 9.1 (2019): 1-9.

3. Heuzé, Céline, et al. "Southern Ocean bottom water characteristics in CMIP5 models." Geophysical Research Letters 40.7 (2013): 1409-1414.

4. Kurtakoti, P., M. Veneziani, A. Stössel, and W. Weijer, 2018: Preconditioning and Formation of Maud Rise Polynyas in a High-Resolution Earth System Model. J. Clim., 31, 9659–9678.

5. Smedsrud, Lars H. "Warming of the deep water in the Weddell Sea along the Greenwich meridian: 1977–2001." Deep Sea Research Part I: Oceanographic Research Papers 52.2 (2005): 241-258.

---

## Editor Comment (EC1) · Matthew Hecht (Editor) · 26 Aug 2020

Dear Authors, having read and considered the two anonymous reviews, and the comment from Ethan Campbell, I believe it unlikely that you'll be able to overcome the major objections that have been raised. If you believe otherwise then I will consider a revised manuscript. But I do think that you should consider withdrawing at this point.

I'm sorry to deliver this assessment. I know that good work has gone into this effort.

Sincerely yours, –Matthew Hecht

---

## Author Comment (AC1) · 24 Sep 2020

**MS-No.:** os-2020-33

Version: Revision

Title: Subsurface Initiation of Deep Convection near Maud Rise

Author(s): René M. van Westen and Henk A. Dijkstra

**Point-by-point reply to reviewer #1**

September 24, 2020

We thank the reviewer for his/her careful reading and for the useful comments on the manuscript. Below is a point-by-point reply with references to figures which are placed at the bottom of the document.

**Major comments:**

1. (...) They also do not investigate several previously suggested significant preconditioning processes that are relevant to the Maud Rise and the Weddell Sea, including the size of the sub-surface heat reservoir created by accumulation of Warm Deep Water, the strength of the Taylor column associated with the Maud Rise seamount, the strength of the impinging flow, and surface salinity anomalies. (...)

**Author's reply:**

The reviewer is correct that additional preconditioning processes can play a role near Maud Rise. In the revision of van Westen and Dijkstra (2020, https://doi.org/10.5194/os-2020-25, in review) we included an analysis of the surface salinity anomalies, SAM index, and Taylor columns. There is no link between the multidecadal preconditioning near Maud Rise and the SAM index in the CESM. There are indeed Taylor columns present near Maud Rise which precondition the region. These Taylor columns are present over the entire simulation period, but cannot explain the multidecadal time scale of polynya formation.

In the revision here, we have focussed more on the salinity anomalies in the mixed layer prior to Maud Rise Polynya (MRP) formation. We find that the surface is slightly saltier prior to MRP formation with a magnitude of 0.07 Psu w.r.t. the time mean (Figure 1c). Part of these anomalies are related to enhanced Ekman upwelling which entrains salt from below the mixed layer depth (Figure 1d). However, these salinity anomalies are too weak to induce convection near the surface.

**Changes in manuscript:**

In the revision we will mention all of the relevant preconditioning processes, referring also to results in van Westen and Dijkstra (2020). Note that the current manuscript mainly focusses on the initiation of convection near Maud Rise and not on the preconditioning, but we agree with the reviewer that this should be placed into context. The surface salinity analysis will be included in the revised results section.

2. In section 3.1, the authors show the accumulation of subsurface heat and salt prior to the onset of the polynyas but fail to relate it to the heat and salt content of a warm and salty water mass known as the Weddell Deep Water (WDW) which lies between 250- 1500m in the Weddell Sea. A necessary but not sufficient condition for long-lasting, consecutive winter polynyas to occur is the heat reservoir at depth or the heat content of WDW (Martinson et al. 1981; Martin et al. 2013; Cheon et al. 2015; Dufour et al. 2017; Kurtakoti et al. 2018). If the main result in the paper is that the convection is initiated subsurface, a thorough and robust analysis of the stratification and water masses in and around Maud Rise is critical to the authors' justification and the readers' understanding.

**Author's reply:**

We analysed the properties (e.g. temperature and salt) of the Weddell Deep Water (WDW) in more detail. The WDW properties and depth profiles reasonable match with that of observations, but the modelled WDW is slightly warmer and more saline compared to observations (Figure 2; note that the CESM simulation had a spin-up period of 150 years). Observed trends of the WDW are also represented in the CESM and there is no spurious build-up of heat as reported in several low-resolution models. The highest temperature and salinity values of the WDW are found prior to polynya formation. This is in agreement with the subsurface static instabilities which develop prior to MRP formation. During MRP formation, the temperature and salinity of the WDW decrease, also seen in observational records.

**Changes in manuscript:**

We will include an analysis of the properties (e.g. temperature and salt)

of the WDW and discuss the relevant trends of these properties. We will also include a comparison with observational results of the WDW.

3. Please define what is subsurface convection. How is it triggered and how is it maintained? Is this convection that only occurs within the WDW and if so how is it initiated and how is this sustained?

**Author's reply:**

There are three different types of oceanic convection (Akitoma 1999, Su et al. 2016): convection by buoyancy loss (type I convection), thermobaric convection (type II convection) and thermobaric cabbeling (type III convection). A nonlinear equation of state is essential only for the latter two types. In type I convection, the mixed layer gradually deepens by the loss of surface buoyancy. Near Maud Rise, surface buoyancy loss occurs at the surface when the surface is strongly cooled during Austral winter time and by the contribution of sea-ice formation (e.g. brine rejection). In the CESM, the mixed layer depth is seasonally varying prior to MRP formation (Figure 4). In most MRP literature, only the negative surface buoyancy (by positive salinity anomalies) is considered. Parcels with a negative buoyancy sink and mix with the layers below and reach their equilibrium buoyancy, this depth marks the mixed layer depth.

However, the build-up of a subsurface heat reservoir (Figure 2) induces buoyancy gain by thermal expansion. Parcels located at subsurface depths experience an upward force and mix with the layers above. These upward moving parcels also reach a certain equilibrium buoyancy (with some overshooting by inertia). The upward motion of parcels at subsurface depths is the definition of subsurface convection in the manuscript. The maximum vertical extent of the subsurface convection can be measured using the depth-integrated buoyancy:

$$B(z) = -\int_{-z_{ref}}^{-z} g \frac{\rho_{\theta}(-z_{ref}) - \rho_{\theta}(z')}{\rho_{\theta}(z')} \, \mathrm{d}z' \tag{1}$$

In the previous version of the manuscript, we confusingly referred to the expression above as CAPE. The large subsurface heat reservoir below Maud Rise maintains the subsurface convection and convection ceases when it is depleted.

Note that subsurface convection plays a dominant role in the initiation of MRP in the CESM during model year 231. Subsurface instabilities (Figure 3) vertically mix heat and salinity towards the surface. The following model year 232, the surface is relatively warm and saline and the surface becomes unstable by buoyancy loss (Figure 4b).

**Changes in manuscript:**

We will not use CAPE anymore in the revised manuscript and we will better specify the type of convection (i.e. buoyancy loss and gain).

4. One of the major scientific weaknesses of this study is the context in which CAPE is used. Ocean CAPE is defined for an ocean column as the maximal Potential Energy that can be released by adiabatic vertical parcel rearrangements, arising from thermobaricity (Su et al. 2016a,b). However, that alone does not indicate the onset of deep convection. Fig 3a. is used to indicate the onset of subsurface convection in earlymid July which is not accurate. Fig 3c clearly shows that the subsurface pattern repeats it- self in the 2 years prior to the formation of the polynyas. This only indicated weakening subsurface stratification and not the onset of subsurface convection which is consistent with the accumulation of WDW heat content. If we could also see the Hovmöller of Temperature and Salinity time series that overlaps the time periods used in 3a and 3c, we can then confidently know for sure if convection does begin in the subsurface as claimed by the authors.

**Author's reply:**

To determine precisely where the vertical mixing occurs near Maud Rise in the CESM, we have re-run model year 231 and written out the daily-averaged vertical diffusivities. Convection can be recognised in climate models by the occurrence of large values to the vertical diffusivity (Figure 3c). We find that the subsurface starts to mix and this mixing becomes stronger prior to MRP formation. In the surface layer, there is also some mixing, but the vertical diffusivity values are a factor 3 smaller compared to that of the subsurface prior to MRP formation (Figure 3d). Using the final depth of the parcels (red curve in Figures 3b,c, buoyancy), we find that below (above) this depth high (low) values of vertical diffusivity are found. This suggests that the maximum vertical extent of the region of large vertical diffusivity can be captured by using the definition of depth-integrated buoyancy. The daily-averaged Hovmöller diagram of vertical diffusivity is similar to the static instability diagram (compare Figure 3a and 3c). There is a clear relationship between the magnitude of static instabilities and the vertical diffusivity (Figure 3d).

As mentioned by the reviewer, there are indeed subsurface static instabilities in model year 229 and 230 (Figure 4a). However, the magnitude of these static instabilities is much smaller compared to the ones in model year 231. For example, when taking the depth-averaged and monthly-averaged static instabilities between 200 - 1000 m depths (Figure 4b), the subsurface static instabilities are a factor 4.8 (August model year 229) and 7.9 (August model year 230) smaller compared to August model year 231. Smaller static instabilities result in less vertical diffusivity which do not overcome the stratification near the surface.

**Changes in manuscript:**

We will include the analysis of the daily-averaged vertical diffusivity in the manuscript. We will demonstrate that the subsurface static instabilities are much larger than the surface static instabilities. A link will made with the WDW and the surface salinity anomalies.

---

## Author Comment (AC2) · 24 Sep 2020

**MS-No.:** os-2020-33

Version: Revision

Title: Subsurface Initiation of Deep Convection near Maud Rise

Author(s): René M. van Westen and Henk A. Dijkstra

**Point-by-point reply to reviewer #2**

September 24, 2020

We thank the reviewer for his/her careful reading and for the useful comments on the manuscript. Below is a point-by-point reply with references to figures which are placed at the bottom of the document.

**Specific Comments:**

1. The growing subsurface instabilities that precede the polynya in CESM model year 231 (Fig. 3a, 3b) are intriguing. It would be great if the authors could discuss in further detail what mechanism is behind the growing instability. Is it related to the multidecadal buildup of subsurface heat, mentioned in pg. 12, L24-25? Or is a shorter-timescale process more relevant here? Pg. 8 would be a good place to discuss this.

**Author's reply:**

We have conducted additional analysis on the Weddell Deep Water (WDW), see Figure 1). During Maud Rise Polynya (MRP) events, the WDW temperature decreases as heat is ventilated by vertical mixing. After the MRP event during model years 205 – 209, the WDW is relatively cold and the temperature increases over time. Prior to model year 231, the WDW temperature reaches maximum values which corresponds to the growing subsurface static instabilities. In model year 231, an MRP forms and the WDW temperature decreases again. The build-up of the WDW heat reservoir has the same dominant period of the multidecadal build-up of subsurface heat over Maud Rise.

**Changes in manuscript:**

We will include an analysis of the properties of the WDW in the revised manuscript. Here we will discuss the multidecadal build-up of subsurface heat is causing the growing subsurface static instabilities. 2. The aforementioned subsurface instabilities also appear to occur in the model years preceding year 231 (Fig. 3c). If these instabilities play a role in initiating convection, why are there no polynyas during this previous time period? Are unfavorable near-surface conditions inhibiting deep convection from above? The time series of wind stress curl, as shown in Fig. 3a, could be useful if it is extended to include this earlier time period.

**Author's reply:**

We have analysed the daily-averaged and monthly-averaged wind-stress curl fields for model years 229, 230 and 231 averaged over the Polynya region. In addition, we determined the wind-stress curl climatology (model years 150 - 250). The climatology shows a persistent negative wind-stress curl over all the months, as well as during model years 229, 230 and 231 (Figure 2a).

We find that during model year 229, the monthly-averaged wind-stress curl is weaker (less negative) w.r.t. climatology during August and October; for September it is slightly stronger (more negative) w.r.t. the climatology mean. For model year 230, the wind-stress curl is weaker w.r.t. the climatology mean during August – October. For August model year 231 (during this month an MRP formed), the wind-stress curl is stronger w.r.t. the climatology mean. On the other hand, while analysing daily-averaged fields (Figure 2b), we find strong negative values for the wind-stress curl during model years 229, 230 and 231. These strong negative values are associated with the passing of (strong) winter cyclones.

Another important effect of MRP formation is the magnitude of the subsurface static instabilities (Figure 3a). There are indeed subsurface instabilities during model years 229 and 230, as pointed out by the reviewer. However, the magnitude of these static instabilities is much smaller compared to the ones in model year 231. For example, when taking the depth-averaged and monthly-averaged static instabilities between 200 – 1000 m depths (Figure 3b), the subsurface static instabilities are a factor 4.8 (August model year 229) and 7.9 (August model year 230) smaller compared to August model year 231. Even with favourable near-surface conditions, such as strong winter cyclones, the subsurface static instabilities are not strong enough to cause convection in the years before model year 231.

**Changes in manuscript:**

We will include the climatology of the wind-stress curl and the monthlyaveraged and daily-averaged results in the revised manuscript. We will discuss why an MRP did not form in the years prior to model year 231, making use of the monthly-averaged and depth-averaged static instabilities. As mentioned in the manuscript, due the time averaging, the magnitude of monthly-averaged subsurface static instabilities are smaller compared to daily-averaged subsurface static instabilities. To demonstrate the growing subsurface static instabilities prior to MRP formation, we will include a time series of the depth-averaged and dailyaveraged subsurface static instabilities for model year 231.

3. Pg. 8, L28-29 and pg. 9, L1-2: Wind stress curl is associated with upwelling, turbulent mixing, and sea-ice divergences. The manuscript would benefit from the authors explicitly quantifying the upwelling magnitude associated with wind stress curl anomalies. For instance, the horizontal and/or vertical Ekman velocities can be inferred from wind stress curl (e.g. Campbell et al., 2019, Methods, salinity fluxes from upwelling). This quantity could help contextualize the near-surface destratification shown in Fig. 3.

**Author's reply:**

We followed the suggestion of the reviewer and determined the salt transport induced by upwelling over the Polynya region, following Campbell et al. (2019). The negative wind-stress curl (cf. Comment 2) entrains salt from below the mixed layer throughout the year. Note that we only determined the salt from upwelling during the sea-ice free months (January – May), the surface drag decreases in the presence of sea ice. The total salinity entrained by Ekman dynamics during the sea-ice free months is 0.065 Psu (time mean model years 150 - 250). For model year 229, 230 and 231 the total salinity (during sea-ice free months) is 0.139, 0.052 and 0.104 Psu, respectively (Figure 2d).

We also determined the salt averaged over the mixed layer depth  $(S_{ml},$  Figure 2c). The  $S_{ml}$  is seasonally varying with a peak-to-peak difference of about 0.6 Psu. We determined the climatology of  $S_{ml}$  over all non-polynya years (note that vertical mixing brings up salt and skews the distribution). The  $S_{ml}$  is increasing between model years 229 –

231. However, the salinity anomalies are not anomalously high and are still within the  $S_{ml}$  climatology ( $\overline{S_{ml}} = 34.42 - 34.50$  Psu, 5 - 95% percentile). The  $S_{ml}$  anomaly (w.r.t. climatology mean) in August model year 231 is 0.07 Psu, the anomaly of salt entrainment by Ekman dynamics is 0.039 Psu.

The salt anomalies could in principle induce convection by surface static instabilities. Therefore, we re-run model year 231 and written out the daily-averaged vertical diffusivities (Figure 4c). We find no increase in the near-surface vertical diffusivity prior to polynya formation, however, the subsurface vertical diffusivity is strongly increasing prior to polynya formation (Figure 4d). Ekman upwelling favours the destratification of the Polynya region, but the induced salt anomalies are simply too weak to induce surface convection in model year 231.

**Changes in manuscript:**

We will include the analysis of Ekman upwelling and discuss the effect of salt anomalies on the destratification over the Polynya region. We will also include the results of the vertical diffusivity demonstrating that the subsurface is responsible for the overturning of the water column in model year 231.

4. When comparing subsurface convection in CESM and Mercator output (Pg. 11, L29-35), it is important to acknowledge the magnitude of ocean heat content variability in the models used. Climate models are known to be prone to excessive subsurface heat accumulation, which has been attributed to freshwater forcing biases (Stössel et al., 2015) and weak parameterized mixing under sea ice (Heuzé et al., 2013). For instance, are subsurface warming trends between the simulated polynya events in CESM (Pg. 2, L24-25) consistent with the observed .032 K/decade trend in Weddell Deep Water temperature between 1977-2001 (Smedsrud, 2005)? What about the Mercator output? If not, the model-to-observation comparison could still be useful, but the difference must be acknowledged to properly inform the interpretation of the data.

**Author's reply:**

We have determined the WDW properties as suggested by the reviewer (Figure 1). There is no excessive subsurface heat accumulation as reported in other studies. Between polynya years (model years 210 - 230), we find a WDW temperature trend of 0.064 K per decade. Between

model years 210 - 240, we find a trend of 0.024 K per decade. These trends are in the same order of Smedsrud (2005). The peak-to-peak difference in the WDW temperature is ~ 0.1°C.

The WDW temperature and salinity values are slightly higher compared to observations by about  $0.1^{\circ}$ C and 0.05 Psu. Note that the CESM had a spin-up period of 150 years. Analysing model years 1 – 10 showed that the WDW (as well as for the WSDW and WSBW) is much better in agreement with observations along the Greenwich meridian.

For the Mercator we also analysed the WDW properties between 1993 – 2018. We find relatively low WDW temperatures after several (small) polynya events, such as in 1996, 2006 and 2018, in agreement with the results presented in Campbell et al. (2019). The peak-to-peak difference in the WDW temperature is about 0.05°C, half of that compared to the CESM. The limited subsurface heat reservoir explains why subsurface convection ceases during 2016.

**Changes in manuscript:**

We will include an analysis of the properties (e.g. temperature and salt) of the WDW and discuss the relevant trends of these properties in the revised paper. We will also include a comparison with observational results of the WDW. Finally, the Mercator analysis will be discussed in the revision.

Figure 1: (a & b): Zonal mean  $(1^{\circ}W - 1^{\circ}E)$  of temperature and salinity along the Greenwich Meridian. The displayed values are yearly averages for model year 215. The black contour displays the 0°C, which marks the WDW. (c & d): Time evolution of the temperature and salinity of the WDW. The dashed lines are linear trends between model years 210 - 230 (red) and model years 210 - 240 (blue). The shading indicates the polynya years.

---

## Author Comment (AC3) · 24 Sep 2020

**MS-No.:** os-2020-33

**Version:** Revision

**Title:** Subsurface Initiation of Deep Convection near Maud Rise

**Author(s):** René M. van Westen and Henk A. Dijkstra

**Point-by-point reply to reviewer #3**

September 24, 2020

We thank Ethan Campbell for his careful reading and for the very useful comments on the manuscript. We have divided the comments into six specific ones and answer those point by point with reference to figures at the bottom of the document.

Specific Comments:

1. *Repeat hydrographic surveys from 1984-2008 along the Greenwich Meridian, crossing Maud Rise, show no sign of marked heat accumulation following ventilation during the massive 1974-1976 Weddell polynyas (Fahrbach et al. 2011). The small warming trend observed is over one order of magnitude smaller than that examined in the authors' companion submission following polynya events (van Westen and Dijkstra, in review), and large decadal fluctuations complicate the detection of even that small observed trend. In fact, local rebound in heat content near Maud Rise following the major 1974-1976 polynyas occurred by 1984 at the latest (Smedsrud 2005), and likely in shorter time. Analysis of about 3,000 temperature profiles from 2002-2017 from ships, floats, and instrumented seals indicated that there was no local buildup of mid-depth heat leading to the 2016-2017 Maud Rise polynyas (Campbell et al. 2019; see section 'Sub-pycnocline temperature records' and Extended Data Fig. 9). These substantial differences between models and reality unfortunately limit the utility of the authors' CESM simulation as a direct analogue for investigating the recent polynya events.*

   **Author's reply:**
   We have examined the properties (e.g. temperature and salinity) of the Weddell Deep Water (WDW) along the Greenwich Meridian (Figure 1).

During Maud Rise Polynya (MRP) events, the WDW temperature decreases as heat is ventilated by vertical mixing. After the MRP event during model years 205 – 209, the WDW is relatively cold and the temperature increases over time. Prior to model year 231, the WDW temperature reaches maximum values which corresponds to the growing subsurface instabilities. In model year 231, an MRP forms and the WDW temperature decreases again. The build-up of the WDW heat reservoir has the same period of the multidecadal build-up of subsurface heat over Maud Rise.

Between polynya years (model years 210 – 230), we find a WDW temperature trend of 0.064 K per decade. Between model years 210 – 240, we find a trend of 0.024 K per decade. These trends are of the same order as in Smedsrud (2005). The peak-to-peak difference in the WDW temperature is $\sim 0.1°$C.

We also analysed the WDW using Mercator (1993 – 2018) along the Greenwich Meridian. We find that in Mercator, prior to the 2016 – 2017 MRP, the WDW is relatively warmer compared to the time mean (1993 – 2018). There is inter-annual variability of the WDW temperature after the appearance of small MRPs with a peak-to-peak difference of $0.05°$, half of that compared to the CESM. The limited subsurface heat reservoir explains why subsurface convection ceases during 2016.

**Changes in manuscript:**
We will include an analysis of the WDW and compare these with observational records; both the CESM and Mercator results will be discussed here.

2. *In contrast, the published high-resolution model simulations that best reproduce the Maud Rise polynya phenomenon point to upper-ocean destratification from surface salinity anomalies, not subsurface heat accumulation, as most important in triggering polynyas (Kurtakoti et al. 2018; Kaufman et al. 2020). These papers should be cited and discussed. Along similar lines, the authors neglect the observational and theoretical evidence that points to low upper-ocean haline stratification as a critical factor in allowing Maud Rise polynyas to emerge in 2016 and 2017 but not in most other years (Campbell et al. 2019). It is inaccurate to characterize our study as attributing the 2016 polynya solely to intense winter storms (as stated in the authors' Abstract,*

*Lines 1-2; Page 8, Lines 21-23; Page 12, Lines 11-12). Both weak upper-ocean stratification and strong storms appear to have been necessary, and strong storms which were more frequent than usual in 2016 but are still a regular occurrence in most (non-polynya) years ? are apparently not a sufficient condition for polynya formation. While the authors attribute the 2017 polynya to a 'weakly stable surface layer' (e.g., Abstract, Line 7), they look elsewhere for the immediate cause of the earlier 2016 polynya. I find this puzzling, as the profiling float measurements examined in Campbell et al. (2019) show that a 'weakly stable surface layer' existed both prior to the 2016 polynya as well as during the following winter.*

**Author's reply:**
We agree with the reviewer that the surface layer was indeed weakly stratified in 2016. We have investigated in more detail the surface salinity anomalies in the CESM (Figure 2). Salt from Ekman upwelling (following Campbell et al. (2019)) shows that more salt is entrained during sea-ice free months (January – May) compared to the time mean (anomaly of 0.039 Psu in model year 231). The depth-averaged salinity over the mixed layer depth was also more saline in model year 231 and the magnitude of these anomalies are comparable, but slightly weaker, compared to Kurtakoti et al. (2018).

To examine the effect of surface driven convection by these salt anomalies, we have re-run the CESM and written out the daily-averaged vertical diffusivities for model year 231 (Figure 3c). The vertical diffusivity can be used to examine vertical mixing in the CESM, as is done in Dufour et al. (2017). We find that the mixed layer displays some vertical mixing, however, the magnitude is a factor 3 smaller compared to the subsurface mixing prior to MRP formation (Figure 3d). Surface salinity anomalies can contribute to preconditioning of the Maud Rise region, but are too weak to induce deep convection.

**Changes in manuscript:**
We will include an analysis of the surface salinity anomalies and the vertical diffusivities in the revised paper. These results will be compared to observations and other model studies.

3. *Surface-driven deep convection is a phenomenon that has been well-documented in decades of observations from the North Atlantic and*

*Mediterranean Sea (e.g., de Jong et al. 2012; Testor et al. 2018). Its theoretical underpinnings are also reasonably well-understood (e.g., Marshall and Schott 1999). In my view, there is little reason to expect that deep convection in the Weddell Sea is not similarly initiated by buoyancy loss and/or dynamic perturbations in the upper ocean. Indeed, this is the canonical model for the formation of polynyas near Maud Rise (Martinson et al. 1981; Motoi et al. 1987). However, if the authors wish to assert that subsurface-initiated deep convection is of real importance, I would suggest critically engaging with literature that may offer a theoretical basis for this mechanism. Harcourt (2005) and Akitomo (2019), for instance, describe subtle processes by which subsurface overturning may be initiated through nonlinearities in the seawater equation of state, but it is important to recognise that this is a fundamentally different sequence of events than that which seems to be occurring in the authors' CESM run.*

**Author's reply:**
There are three different types of oceanic convection (Akitoma 1999, Su et al. 2016): convection by buoyancy loss (type I convection), thermobaric convection (type II convection) and thermobaric cabbeling (type III convection). A nonlinear equation of state is essential only for the latter two types. We are dealing here with just type I convection.

Near Maud Rise, surface buoyancy loss occurs at the surface when the surface is strongly cooled during Austral winter time and by the contribution of sea-ice formation (e.g. brine rejection). In the CESM, the mixed layer depth is seasonally varying prior to MRP formation (Figure 4). The build-up of a subsurface heat reservoir (Figure 1) induces buoyancy gain by thermal expansion. Parcels located at subsurface depths experience an upward force and mix with the layers above.

To study the subsurface mixing, we have written out the daily-averaged vertical diffusivities (Figure 3c) from the CESM simulation. The subsurface vertical diffusivities are increasing indicating that the water column overturns at subsurface depths. There is no increase in the surface vertical diffusivities prior to MRP formation (Figure 3d).

**Changes in manuscript:**
We will include a better description of the subsurface convection and

the analysis of the vertical diffusivities in the revised manuscript.

4. *Regardless of the theoretical feasibility of subsurface-initiated overturning, the phenomenon that the authors highlight from their CESM run is unlikely to have occurred in 2016. In the attached Figure 1 (see below), I have plotted potential density profiles from the Argo float 5904471, which was present at Maud Rise during the recent polynyas (Campbell et al. 2019). This figure is a direct comparison to the authors' Figure 2c from their CESM run, which shows a statically unstable profile below about 75 m prior to polynya formation. In contrast, the observations show that potential density increased steadily with depth, including immediately prior to the 2016 polynya, as is typical throughout the world ocean. While the CESM profile seems to be characterized by a bolus of mid-depth heat anomalies, which create a remarkably thick potential density inversion layer, these features are unsurprisingly absent in the observations. (This is to say nothing of the model's twofold bias in its deep-to-surface potential density difference, which is greater than 0.2 kg/m3 in the model but is no more than 0.1 kg/m3 in the observations, or the inappropriate use of surface-referenced potential density [sigma-theta] to characterize inversions when a locally-referenced potential density should be used instead.)*

**Author's reply:**
As stated in the manuscript, subsurface static instabilities are (very) localised in the CESM. The static instabilities are found below the regions where the sea ice opens in August for both CESM and Mercator. The Argo float was located outside these static unstable regions and, as expected, the water column is statically stable as shown by the reviewer. This float can therefore not reject our hypothesis and to our knowledge, there are no direct measurements of the water column directly below the region of a developing MRP. To analyse the water column directly below the polynya, we make use of the Mercator. Of course, Mercator also parametrises different processes but the background density profile is comparable to that of the Argo float.

We have analysed the vertical diffusivity in the CESM and these are linked to the growing subsurface static instabilities. These subsurface static instabilities are connected to relatively high temperatures of the WDW. The Mercator diffusivities are not available, but static

instabilities are also found below the region of relatively low sea-ice concentrations over Maud Rise.

**Changes in manuscript:**
Based on this comment, we will discuss the Argo float results in more detail.

5. *The authors seem to dismiss the possibility of using the float observations to conduct an analysis similar to their assessment of CESM model output and Mercator reanalysis data, stating that 'these Argo float observations are too sparse (every 10 days) to analyse the oceanic state (and e.g. convection)' (Page 9, Lines 28-30). In the context of investigating preconditioning for a polynya, this is inaccurate. The oceanic state – particularly below the mixed layer – does not vary substantially on time scales less than 10 days, as seems to be indicated by the authors' own analysis of the CESM output and Mercator data. The float data are plenty useful and are, in fact, the best records we have on conditions in 2016 and 2017 at Maud Rise. (It is important to note here that the Mercator reanalysis data is poorly constrained in the ice-covered Weddell Sea, and should be approached with greater caution than is done here. Comparison with a single float profile (Figure 4b) towards which the reanalysis is likely nudged – is not an adequate validation of its skill.*

**Author's reply:**
In case of convection, the oceanic state can vary on short time scales less than 10 days. Besides, the Argo float was not located inside the region of static instabilities, as discussed in Comment 4. As stated in the manuscript (page 11, lines 10 – 13), we determined the subsurface convection using the Argo float. We find that between 110 – 240 m depths, the water column is statically unstable in the Argo profile. Mercator displays a well-mixed layer between these depths, indicating that the water column has likely overturned at these depths. Note that the Argo float measures at a particular time and the Mercator consists of daily-averaged fields. Although the Argo floats are near Maud Rise, they didn't measure the water column directly below an MRP.

We agree with the reviewer that the Mercator reanalysis products also has its shortcomings, but still reasonable agrees with Argo float and SSMR-SSM/I measurements. The Mercator indicates several interesting results which can not be captured by the Argo float. For example,

the decreased vertical extent of subsurface convection (red curve in Figure 4d) which is a plausible explanation why the MRP closes in 2016. There is no deepening of the mixed layer depth, which is important in the case of surface-driven convection.

**Changes in manuscript:**
We will stress that the Argo floats are not located in the regions of subsurface static instabilities and that there are no other direct observations of the initiation of (sub)surface convection. This also motivates the use of other products such as Mercator in this study.

6. *Ultimately, the float observations analyzed in Campbell et al. (2019) as well as the body of previous work on Weddell Sea polynyas and hydrography offer ample evidence that model-observation disagreement is severe in this region and in the context of this phenomenon. With this in mind, I would argue that it is probably counterproductive to interrogate the causes of observed polynya events through direct comparison with a model that behaves very differently than the real world.*

**Author's reply:**
We agree with the author that also the CESM has its shortcomings, but we do not think the model-data mismatch disqualifies the model for analysing the formation mechanism of MRP events. Note that also the study in Kurtakoti et al. (2018), which the reviewer appears to find credible, is done with only a slightly different CESM configuration.

**Changes in manuscript:**
A discussion on whether the model is fit for purpose, based on a comparison with available observations, will be included in the revised paper.

[Figure]

Figure 1: (a & b): Zonal mean (1°W – 1°E) of temperature and salinity along the Greenwich Meridian. The displayed values are yearly averages for model year 215. The black contour displays the 0°C, which marks the WDW. (c & d): Time evolution of the temperature and salinity of the WDW. The dashed lines are linear trends between model years 210 – 230 (red) and model years 210 – 240 (blue). The shading indicates the polynya years.

[Figure]

Figure 2: (a): Monthly-averaged magnitude of the wind-stress curl over the Polynya region for model years 229, 230 and 231. The shading indicates the climate variability, where the dark (light) shading correspond to the 25% – 75% (5% – 95%) percentile levels, the black curve is the climatology mean (model years 150 – 250). (b): Same as a), but now for daily averages. The shading is the same as in a). (c): Depth-averaged salinity over the mixed layer depth for model years 229, 230 and 231. The shading indicates the climate variability of this quantity (same as a)), but now only the non-MRP years are considered in the climatology. (d): The entrained salinity into the mixed layer from Ekman dynamics for model year 229, 230 and 231. The shading indicates the climate variability of this quantity (same as a)).

[Figure]

Figure 3: (a): Area per depth level where the water column in the Polynya region is statically unstable ($\mathcal{S} > 0$), normalised to the total area at that depth level. The blue curve is the daily-averaged magnitude of the wind-stress curl over the Polynya region, same as in Figure 2b. (b): Final depth of fluid elements due to (subsurface) convection. Only regions where deviations from the initial depth occur are displayed. The red curve shows the final depth of convection initially starting at $z_{ref} = 580$ m. The blue curve (bottom part) is the total area of the model year 231 MRP. (c): The vertical diffusivity ($\chi$) for temperature, spatially averaged over the Polynya region. The red curve is the same as in b). In a) – c), the black curve shows the maximum mixed layer depth spatially averaged over the Polynya region. (d): The depth-averaged vertical diffusivity ($\chi$) and normalised area of static instabilities over the upper 100 m and between 200 – 1000 m. In all figures, the dashed lines indicate the formation (20 August) and ending (14 December) of the MRP.

[Figure]

[Figure]

Figure 4: (a): Area per depth level where the water column in the Polynya region is statically unstable ($\mathcal{S} > 0$), normalised to the total area at that depth level. (b): The depth-averaged normalised area of static instabilities over the upper 100 m and between 200 – 1000 m. In both figures, the dashed lines indicate the beginning (model year 231) and ending (model year 237) of the multiyear MRPs. The model output analysed consists of monthly averages.

---

## Author Comment (AC4) · 24 Sep 2020

Dear Editor, dear Matthew,

Thank you for your assessment of this manuscript. We found the suggestions and the comments of the reviewers helpful. Based on their comments, we performed additional analyses on the CESM simulation. These additional results, in particular those for the daily-averaged vertical diffusivity, further support our hypothesis that subsurface convection plays an important role in MRP formation. We think that these additional findings should be at least reported and hopefully they will convince the reviewers.

[Figure]

We have written a point-by-point reply to the two anonymous reviewers and to Ethan Campbell. We kindly ask you to allow us to submit a revised version of the manuscript.

Sincerely,

René van Westen and Henk Dijkstra
* * *